# Tying Word Vectors and Word Classifiers: A Loss Framework for Language Modeling

**Hakan Inan, Khashayar Khosravi**
Stanford University
Stanford, CA, USA
{inanh,khosravi}@stanford.edu

**Richard Socher**
Salesforce Research
Palo Alto, CA, USA
rsocher@salesforce.com

## ABSTRACT

Recurrent neural networks have been very successful at predicting sequences of words in tasks such as language modeling. However, all such models are based on the conventional classification framework, where the model is trained against one-hot targets, and each word is represented both as an input and as an output in isolation. This causes inefficiencies in learning both in terms of utilizing all of the information and in terms of the number of parameters needed to train. We introduce a novel theoretical framework that facilitates better learning in language modeling, and show that our framework leads to tying together the input embedding and the output projection matrices, greatly reducing the number of trainable variables. Our framework leads to state of the art performance on the Penn Treebank with a variety of network models.

## 1 INTRODUCTION

Neural network models have recently made tremendous progress in a variety of NLP applications such as speech recognition (Irie et al., 2016), sentiment analysis (Socher et al., 2013), text summarization (Rush et al., 2015; Nallapati et al., 2016), and machine translation (Firat et al., 2016).

Despite the overwhelming success achieved by recurrent neural networks in modeling long range dependencies between words, current recurrent neural network language models (RNNLM) are based on the conventional classification framework, which has two major drawbacks: First, there is no assumed metric on the output classes, whereas there is evidence suggesting that learning is improved when one can define a natural metric on the output space (Frogner et al., 2015). In language modeling, there is a well established metric space for the outputs (words in the language) based on word embeddings, with meaningful distances between words (Mikolov et al., 2013; Pennington et al., 2014). Second, in the classical framework, inputs and outputs are considered as isolated entities with no semantic link between them. This is clearly not the case for language modeling, where inputs and outputs in fact live in identical spaces. Therefore, even for models with moderately sized vocabularies, the classical framework could be a vast source of inefficiency in terms of the number of variables in the model, and in terms of utilizing the information gathered by different parts of the model (e.g. inputs and outputs).

In this work, we introduce a novel loss framework for language modeling to remedy the above two problems. Our framework is comprised of two closely linked improvements. First, we augment the classical cross-entropy loss with an additional term which minimizes the KL-divergence between the model's prediction and an estimated target distribution based on the word embeddings space. This estimated distribution uses knowledge of word vector similarity. We then theoretically analyze this loss, and this leads to a second and synergistic improvement: tying together two large matrices by reusing the input word embedding matrix as the output classification matrix. We empirically validate our theory in a practical setting, with much milder assumptions than those in theory. We also find empirically that for large networks, most of the improvement could be achieved by only reusing the word embeddings.

We test our framework by performing extensive experiments on the Penn Treebank corpus, a dataset widely used for benchmarking language models (Mikolov et al., 2010; Merity et al., 2016). We demonstrate that models trained using our proposed framework significantly outperform models

trained using the conventional framework. We also perform experiments on the newly introduced Wikitext-2 dataset (Merity et al., 2016), and verify that the empirical performance of our proposed framework is consistent across different datasets.

## 2    BACKGROUND: RECURRENT NEURAL NETWORK LANGUAGE MODEL

In any variant of recurrent neural network language model (RNNLM), the goal is to predict the next word indexed by $t$ in a sequence of one-hot word tokens $(y_1^*, \ldots y_N^*)$ as follows:

$$x_t = Ly_{t-1}^*, \tag{2.1}$$

$$h_t = f(x_t, h_{t-1}), \tag{2.2}$$

$$y_t = \text{softmax}\left(Wh_t + b\right). \tag{2.3}$$

The matrix $L \in \mathbb{R}^{d_x \times |V|}$ is the word embedding matrix, where $d_x$ is the word embedding dimension and $|V|$ is the size of the vocabulary. The function $f(.,.)$ represents the recurrent neural network which takes in the current input and the previous hidden state and produces the next hidden state. $W \in \mathbb{R}^{|V| \times d_h}$ and $b \in \mathbb{R}^{|V|}$ are the the output projection matrix and the bias, respectively, and $d_h$ is the size of the RNN hidden state. The $|V|$ dimensional $y_t$ models the discrete probability distribution for the next word.

Note that the above formulation does not make any assumptions about the specifics of the recurrent neural units, and $f$ could be replaced with a standard recurrent unit, a gated recurrent unit (GRU) (Cho et al., 2014), a long-short term memory (LSTM) unit (Hochreiter & Schmidhuber, 1997), etc. For our experiments, we use LSTM units with two layers.

Given $y_t$ for the $t^{\text{th}}$ example, a loss is calculated for that example. The loss used in the RNNLMs is almost exclusively the cross-entropy between $y_t$ and the observed one-hot word token, $y_t^*$:

$$J_t = \text{CE}(y_t^* \parallel y_t) = -\sum_{i \in |V|} y_{t,i}^* \log y_{t,i}. \tag{2.4}$$

We shall refer to $y_t$ as the model prediction distribution for the $t^{\text{th}}$ example, and $y_t^*$ as the empirical target distribution (both are in fact conditional distributions given the history). Since cross-entropy and *Kullback-Leibler* divergence are equivalent when the target distribution is one-hot, we can rewrite the loss for the $t^{\text{th}}$ example as

$$J_t = \text{D}_{KL}(y_t^* \parallel y_t). \tag{2.5}$$

Therefore, we can think of the optimization of the conventional loss in an RNNLM as trying to minimize the distance[1] between the model prediction distribution ($y$) and the empirical target distribution ($y^*$), which, with many training examples, will get close to minimizing distance to the actual target distribution. In the framework which we will introduce, we utilize *Kullback-Leibler* divergence as opposed to cross-entropy due to its intuitive interpretation as a distance between distributions, although the two are not equivalent in our framework.

## 3    AUGMENTING THE CROSS-ENTROPY LOSS

We propose to augment the conventional cross-entropy loss with an additional loss term as follows:

$$\hat{y}_t = \text{softmax}\left(Wh_t/\tau\right), \tag{3.1}$$

$$J_t^{aug} = \text{D}_{KL}(\tilde{y}_t \parallel \hat{y}_t), \tag{3.2}$$

$$J_t^{tot} = J_t + \alpha J_t^{aug}. \tag{3.3}$$

In above, $\alpha$ is a hyperparameter to be adjusted, and $\hat{y}_t$ is almost identical to the regular model prediction distribution $y_t$ with the exception that the logits are divided by a temperature parameter $\tau$. We define $\tilde{y}_t$ as some probability distribution that estimates the true data distribution (conditioned on the word history) which satisfies $\mathbb{E}\tilde{y}_t = \mathbb{E}y_t^*$. The goal of this framework is to minimize the

---

[1]We note, however, that *Kullback-Leibler* divergence is not a valid distance metric.

distribution distance between the prediction distribution and a more accurate estimate of the true data distribution.

To understand the effect of optimizing in this setting, let's focus on an ideal case in which we are given the true data distribution so that $\tilde{y}_t = \mathbb{E}y_t^*$, and we only use the augmented loss, $J^{aug}$. We will carry out our investigation through stochastic gradient descent, which is the technique dominantly used for training neural networks. The gradient of $J_t^{aug}$ with respect to the logits $Wh_t$ is

$$\nabla J_t^{aug} = \frac{1}{\tau}(\hat{y}_t - \tilde{y}_t). \tag{3.4}$$

Let's denote by $e_j \in \mathbb{R}^{|V|}$ the vector whose $j^{\text{th}}$ entry is 1, and others are zero. We can then rewrite (3.4) as

$$\tau \nabla J_t^{aug} = \hat{y}_t - \left[ e_1, \ldots, e_{|V|} \right] \tilde{y}_t = \sum_{i \in V} \tilde{y}_{t,i}(\hat{y}_t - e_i). \tag{3.5}$$

Implication of (3.5) is the following: Every time the optimizer sees one training example, it takes a step not only on account of the label seen, but it proceeds taking into account all the class labels for which the conditional probability is not zero, and the relative step size for each step is given by the conditional probability for that label, $\tilde{y}_{t,i}$. Furthermore, this is a much less noisy update since the target distribution is exact and deterministic. Therefore, unless all the examples exclusively belong to a specific class with probability 1, the optimization will act much differently and train with greatly improved supervision.

The idea proposed in the recent work by Hinton et al. (2015) might be considered as an application of this framework, where they try to obtain a good set of $\tilde{y}$'s by training very large models and using the model prediction distributions of those.

Although finding a good $\tilde{y}$ in general is rather nontrivial, in the context of language modeling we can hope to achieve this by exploiting the inherent metric space of classes encoded into the model, namely the space of word embeddings. Specifically, we propose the following for $\tilde{y}$:

$$u_t = Ly_t^*, \tag{3.6}$$

$$\tilde{y}_t = \text{softmax}\left(\frac{L^T u_t}{\tau}\right). \tag{3.7}$$

In words, we first find the target word vector which corresponds to the target word token (resulting in $u_t$), and then take the inner product of the target word vector with all the other word vectors to get an unnormalized probability distribution. We adjust this with the same temperature parameter $\tau$ used for obtaining $\hat{y}_t$ and apply softmax. The target distribution estimate, $\tilde{y}$, therefore measures the similarity between the word vectors and assigns similar probability masses to words that the language model deems close. Note that the estimation of $\tilde{y}$ with this procedure is iterative, and the estimates of $\tilde{y}$ in the initial phase of the training are not necessarily informative. However, as training procedes, we expect $\tilde{y}$ to capture the word statistics better and yield a consistently more accurate estimate of the true data distribution.

## 4 THEORETICALLY DRIVEN REUSE OF WORD EMBEDDINGS

We now theoretically motivate and introduce a second modification to improve learning in the language model. We do this by analyzing the proposed augmented loss in a particular setting, and observe an implicit core mechanism of this loss. We then make our proposition by making this mechanism explicit.

We start by introducing our setting for the analysis. We restrict our attention to the case where the input embedding dimension is equal to the dimension of the RNN hidden state, i.e. $d \triangleq d_x = d_h$. We also set $b = 0$ in (2.3) so that $y_t = Wh_t$. We only use the augmented loss, i.e. $J^{tot} = J^{aug}$, and we assume that we can achieve zero training loss. Finally, we set the temperature parameter $\tau$ to be large.

We first show that when the temperature parameter, $\tau$, is high enough, $J_t^{aug}$ acts to match the logits of the prediction distribution to the logits of the the more informative labels, $\tilde{y}$. We proceed in the same

way as was done in Hinton et al. (2015) to make an identical argument. Particularly, we consider the derivative of $J_t^{aug}$ with respect to the entries of the logits produced by the neural network.

Let's denote by $l_i$ the $i^{\text{th}}$ column of L. Using the first order approximation of exponential function around zero ($\exp(x) \approx 1 + x$), we can approximate $\tilde{y}_t$ (same holds for $\hat{y}_t$) at high temperatures as follows:

$$\tilde{y}_{t,i} = \frac{\exp\left(\langle u_t, l_i \rangle / \tau\right)}{\sum_{j \in V} \exp\left(\langle u_t, l_j \rangle / \tau\right)} \approx \frac{1 + \langle u_t, l_i \rangle / \tau}{|V| + \sum_{j \in V} \langle u_t, l_j \rangle / \tau}. \tag{4.1}$$

We can further simplify (4.1) if we assume that $\langle u_t, l_j \rangle = 0$ on average:

$$\tilde{y}_{t,i} \approx \frac{1 + \langle u_t, l_i \rangle / \tau}{|V|}. \tag{4.2}$$

By replacing $\tilde{y}_t$ and $\hat{y}_t$ in (3.4) with their simplified forms according to (4.2), we get

$$\frac{\partial J_t^{aug}}{\partial (Wh_t)_i} \rightarrow \frac{1}{\tau^2 |V|} \left(Wh_t - L^T u_t\right)_i, \quad \text{as } \tau \rightarrow \infty, \tag{4.3}$$

which is the desired result that augmented loss tries to match the logits of the model to the logits of $\tilde{y}$'s. Since the training loss is zero by assumption, we necessarily have

$$Wh_t = L^T u_t \tag{4.4}$$

for each training example, i.e., gradient contributed by each example is zero. Provided that $W$ and $L$ are full rank matrices and there are more linearly independent examples of $h_t$'s than the embedding dimension $d$, we get that the space spanned by the columns of $L^T$ is equivalent to that spanned by the columns of $W$. Let's now introduce a square matrix $A$ such that $W = L^T A$. (We know $A$ exists since $L^T$ and $W$ span the same column space). In this case, we can rewrite

$$Wh_t = L^T Ah_t \triangleq L^T \tilde{h}_t. \tag{4.5}$$

In other words, by reusing the embedding matrix in the output projection layer (with a transpose) and letting the neural network do the necessary linear mapping $h \rightarrow Ah$, we get the same result as we would have in the first place.

Although the above scenario could be difficult to exactly replicate in practice, it uncovers a mechanism through which our proposed loss augmentation acts, which is trying to constrain the output (unnormalized) probability space to a small subspace governed by the embedding matrix. This suggests that we can make this mechanism explicit and constrain $W = L^T$ during training while setting the output bias, $b$, to zero. Doing so would not only eliminate a big matrix which dominates the network size for models with even moderately sized vocabularies, but it would also be optimal in our setting of loss augmentation as it would eliminate much work to be done by the augmented loss.

## 5 RELATED WORK

Since their introduction in Mikolov et al. (2010), many improvements have been proposed for RNNLMs , including different dropout methods (Zaremba et al., 2014; Gal, 2015), novel recurrent units (Zilly et al., 2016), and use of pointer networks to complement the recurrent neural network (Merity et al., 2016). However, none of the improvements dealt with the loss structure, and to the best of our knowledge, our work is the first to offer a new loss framework.

Our technique is closely related to the one in Hinton et al. (2015), where they also try to estimate a more informed data distribution and augment the conventional loss with KL divergence between model prediction distribution and the estimated data distribution. However, they estimate their data distribution by training large networks on the data and then use it to improve learning in smaller networks. This is fundamentally different from our approach, where we improve learning by transferring knowledge between different parts of the same network, in a self contained manner.

The work we present in this paper is based on a report which was made public in Inan & Khosravi (2016). We have recently come across a concurrent preprint (Press & Wolf, 2016) where the authors reuse the word embedding matrix in the output projection to improve language modeling.

However, their work is purely empirical, and they do not provide any theoretical justification for their approach. Finally, we would like to note that the idea of using the same representation for input and output words has been explored in the past, and there exists language models which could be interpreted as simple neural networks with shared input and output embeddings (Bengio et al., 2001; Mnih & Hinton, 2007). However, shared input and output representations were implicitly built into these models, rather than proposed as a supplement to a baseline. Consequently, possibility of improvement was not particularly pursued by sharing input and output representations.

# 6 EXPERIMENTS

In our experiments, we use the Penn Treebank corpus (PTB) (Marcus et al., 1993), and the Wikitext-2 dataset (Merity et al., 2016). PTB has been a standard dataset used for benchmarking language models. It consists of 923k training, 73k validation, and 82k test words. The version of this dataset which we use is the one processed in Mikolov et al. (2010), with the most frequent 10k words selected to be in the vocabulary and rest replaced with a an <unk> token [2]. Wikitext-2 is a dataset released recently as an alternative to PTB[3]. It contains $2,088$k training, $217$k validation, and $245$k test tokens, and has a vocabulary of $33,278$ words; therefore, in comparison to PTB, it is roughly 2 times larger in dataset size, and 3 times larger in vocabulary.

## 6.1 MODEL AND TRAINING HIGHLIGHTS

We closely follow the LSTM based language model proposed in Zaremba et al. (2014) for constructing our baseline model. Specifically, we use a 2-layer LSTM with the same number of hidden units in each layer, and we use 3 different network sizes: small (200 units), medium (650 units), and large (1500 units). We train our models using stochastic gradient descent, and we use a variant of the dropout method proposed in Gal (2015). We defer further details regarding training the models to section A of the appendix. We refer to our baseline network as variational dropout LSTM, or VD-LSTM in short.

## 6.2 EMPIRICAL VALIDATION FOR THE THEORY OF REUSING WORD EMBEDDINGS

In Section 4, we showed that the particular loss augmentation scheme we choose constrains the output projection matrix to be close to the input embedding matrix, without explicitly doing so by reusing the input embedding matrix. As a first experiment, we set out to validate this theoretical result. To do this, we try to simulate the setting in Section 4 by doing the following: We select a randomly chosen $20,000$ contiguous word sequence in the PTB training set, and train a 2-layer LSTM language model with 300 units in each layer with loss augmentation by minimizing the following loss:

$$J^{tot} = \beta J^{aug}\tau^2|V| + (1 - \beta)J. \tag{6.1}$$

Here, $\beta$ is the proportion of the augmented loss used in the total loss, and $J^{aug}$ is scaled by $\tau^2|V|$ to approximately match the magnitudes of the derivatives of $J$ and $J^{aug}$ (see (4.3)). Since we aim to achieve the minimum training loss possible, and the goal is to show a particular result rather than to achieve good generalization, we do not use any kind of regularization in the neural network (e.g. weight decay, dropout). For this set of experiments, we also constrain each row of the input embedding matrix to have a norm of 1 because training becomes difficult without this constraint when only augmented loss is used. After training, we compute a metric that measures distance between the subspace spanned by the rows of the input embedding matrix, $L$, and that spanned by the columns of the output projection matrix, $W$. For this, we use a common metric based on the relative residual norm from projection of one matrix onto another (Björck & Golub, 1973). The computed distance between the subspaces is 1 when they are orthogonal, and 0 when they are the same. Interested reader may refer to section B in the appendix for the details of this metric.

Figure 1 shows the results from two tests. In one (panel a), we test the effect of using the augmented loss by sweeping $\beta$ in (6.1) from 0 to 1 at a reasonably high temperature ($\tau = 10$). With no loss

---

[2]PTB can be downloaded at http://www.fit.vutbr.cz/ imikolov/rnnlm/simple-examples.tgz
[3]Wikitext-2 can be downloaded at https://s3.amazonaws.com/research.metamind.io/wikitext/wikitext-2-v1.zip

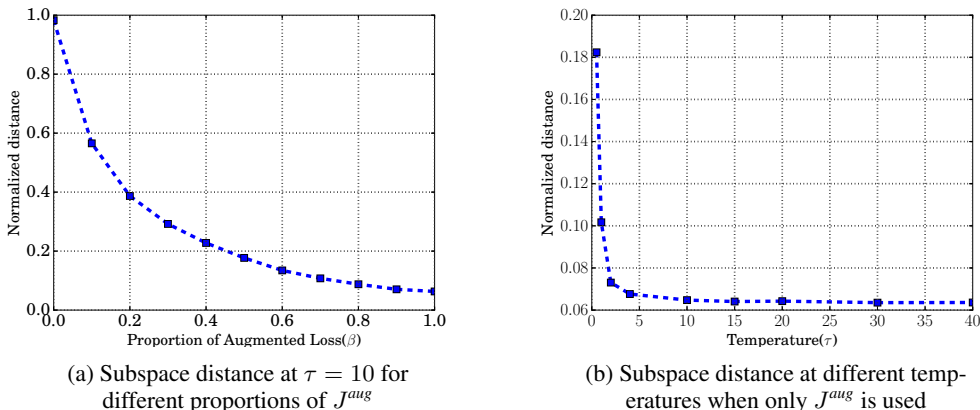

(a) Subspace distance at $\tau = 10$ for different proportions of $J^{aug}$

(b) Subspace distance at different temperatures when only $J^{aug}$ is used

Figure 1: Subspace distance between $L^T$ and $W$ for different experiment conditions for the validation experiments. Results are averaged over 10 independent runs. These results validate our theory under practical conditions.

augmentation ($\beta = 0$), the distance is almost 1, and as more and more augmented loss is used the distance decreases rapidly, and eventually reaches around 0.06 when only augmented loss is used. In the second test (panel b), we set $\beta = 1$, and try to see the effect of the temperature on the subspace distance (remember the theory predicts low distance when $\tau \to \infty$). Notably, the augmented loss causes $W$ to approach $L^T$ sufficiently even at temperatures as low as 2, although higher temperatures still lead to smaller subspace distances.

These results confirm the mechanism through which our proposed loss pushes $W$ to learn the same column space as $L^T$, and it suggests that reusing the input embedding matrix by explicitly constraining $W = L^T$ is not simply a kind of regularization, but is in fact an optimal choice in our framework. What can be achieved separately with each of the two proposed improvements as well as with the two of them combined is a question of empirical nature, which we investigate in the next section.

## 6.3 RESULTS ON PTB AND WIKITEXT-2 DATASETS

In order to investigate the extent to which each of our proposed improvements helps with learning, we train 4 different models for each network size: (1) 2-Layer LSTM with variational dropout (VD-LSTM) (2) 2-Layer LSTM with variational dropout and augmented loss (VD-LSTM +AL) (3) 2-Layer LSTM with variational dropout and reused embeddings (VD-LSTM +RE) (4) 2-Layer LSTM with variational dropout and both RE and AL (VD-LSTM +REAL).

Figure 2 shows the validation perplexities of the four models during training on the PTB corpus for small (panel a) and large (panel b) networks. All of AL, RE, and REAL networks significantly outperform the baseline in both cases. Table 1 compares the final validation and test perplexities of the four models on both PTB and Wikitext-2 for each network size. In both datasets, both AL and RE improve upon the baseline individually, and using RE and AL together leads to the best performance. Based on performance comparisons, we make the following notes on the two proposed improvements:

- AL provides better performance gains for smaller networks. This is not surprising given the fact that small models are rather inflexible, and one would expect to see improved learning by training against a more informative data distribution (contributed by the augmented loss) (see Hinton et al. (2015)). For the smaller PTB dataset, performance with AL surpasses that with RE. In comparison, for the larger Wikitext-2 dataset, improvement by AL is more limited. This is expected given larger training sets better represent the true data distribution, mitigating the supervision problem. In fact, we set out to validate this reasoning in a direct manner, and additionally train the small networks separately on the first and second halves of the Wikitext-2 training set. This results in two distinct datasets which

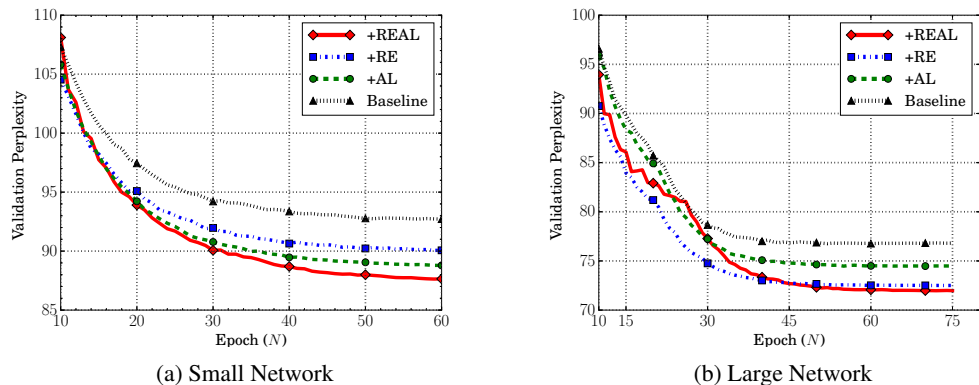

(a) Small Network (b) Large Network

Figure 2: Progress of validation perplexities during training for the 4 different models for two (small (200) and large (1500)) network sizes.

are each about the same size as PTB (1044K vs 929K). As can be seen in Table 2, AL has significantly improved competitive performance against RE and REAL despite the fact that embedding size is 3 times larger compared to PTB. These results support our argument that the proposed augmented loss term acts to improve the amount of information gathered from the dataset.

- RE significantly outperforms AL for larger networks. This indicates that, for large models, the more effective mechanism of our proposed framework is the one which enforces proximity between the output projection space and the input embedding space. From a model complexity perspective, the nontrivial gains offered by RE for all network sizes and for both datasets could be largely attributed to its explicit function to reduce the model size while preserving the representational power according to our framework.

We list in Table 3 the comparison of models with and without our proposed modifications on the Penn Treebank Corpus. The best LSTM model (VD-LSTM+REAL) outperforms all previous work which uses conventional framework, including large ensembles. The recently proposed recurrent highway networks (Zilly et al., 2016) when trained with reused embeddings (VD-RHN +RE) achieves the best overall performance, improving on VD-RHN by a perplexity of 2.5.

Table 1: Comparison of the final word level perplexities on the validation and test set for the 4 different models.

| Network | Model | PTB | | Wikitext-2 | |
| --- | --- | --- | --- | --- | --- |
| | | Valid | Test | Valid | Test |
| Small[4] (200 units) | VD-LSTM | 92.6 | 87.3 | 112.2 | 105.9 |
| | VD-LSTM+AL | 86.3 | 82.9 | 110.3 | 103.8 |
| | VD-LSTM+RE | 89.9 | 85.1 | 106.1 | 100.5 |
| | VD-LSTM+REAL | 86.3 | 82.7 | 105.6 | 98.9 |
| Medium (650 units) | VD-LSTM | 82.0 | 77.7 | 100.2 | 95.3 |
| | VD-LSTM+AL | 77.4 | 74.7 | 98.8 | 93.1 |
| | VD-LSTM+RE | 77.1 | 73.9 | 92.3 | 87.7 |
| | VD-LSTM+REAL | 75.7 | 73.2 | 91.5 | 87.0 |
| Large[5] (1500 units) | VD-LSTM | 76.8 | 72.6 | - | - |
| | VD-LSTM+AL | 74.5 | 71.2 | - | - |
| | VD-LSTM+RE | 72.5 | 69.0 | - | - |
| | VD-LSTM+REAL | 71.1 | 68.5 | - | - |

Table 2: Performance of the four different small models trained on the equally sized two partitions of Wikitext-2 training set. These results are consistent with those on PTB (see Table 1), which has a similar training set size with each of these partitions, although its word embedding dimension is three times smaller.

| Network | Model | Wikitext-2, Partition 1 | | Wikitext-2, Partition 2 | |
|---|---|---|---|---|---|
| | | Valid | Test | Valid | Test |
| Small (200 units) | VD-LSTM | 159.1 | 148.0 | 163.19 | 148.6 |
| | VD-LSTM+AL | 153.0 | 142.5 | 156.4 | 143.7 |
| | VD-LSTM+RE | 152.4 | 141.9 | 152.5 | 140.9 |
| | VD-LSTM+REAL | 149.3 | 140.6 | 150.5 | 138.4 |

Table 3: Comparison of our work to previous state of the art on word-level validation and test perplexities on the Penn Treebank corpus. Models using our framework significantly outperform other models.

| Model | Parameters | Validation | Test |
|---|---|---|---|
| RNN (Mikolov & Zweig) | 6M | - | 124.7 |
| RNN+LDA (Mikolov & Zweig) | 7M | - | 113.7 |
| RNN+LDA+KN-5+Cache (Mikolov & Zweig) | 9M | - | 92.0 |
| Deep RNN (Pascanu et al., 2013a) | 6M | - | 107.5 |
| Sum-Prod Net (Cheng et al., 2014) | 5M | - | 100.0 |
| LSTM (medium) (Zaremba et al., 2014) | 20M | 86.2 | 82.7 |
| CharCNN (Kim et al., 2015) | 19M | - | 78.9 |
| LSTM (large) (Zaremba et al., 2014) | 66M | 82.2 | 78.4 |
| VD-LSTM (large, untied, MC) (Gal, 2015) | 66M | - | $73.4 \pm 0.0$ |
| Pointer Sentinel-LSTM(medium) (Merity et al., 2016) | 21M | 72.4 | 70.9 |
| 38 Large LSTMs (Zaremba et al., 2014) | 2.51B | 71.9 | 68.7 |
| 10 Large VD-LSTMs (Gal, 2015) | 660M | - | 68.7 |
| VD-RHN (Zilly et al., 2016) | 32M | 71.2 | 68.5 |
| VD-LSTM +REAL (large) | 51M | 71.1 | 68.5 |
| VD-RHN +RE (Zilly et al., 2016) [6] | 24M | **68.1** | **66.0** |

## 6.4 QUALITATIVE RESULTS

One important feature of our framework that leads to better word predictions is the explicit mechanism to assign probabilities to words not merely according to the observed output statistics, but also considering the metric similarity between words. We observe direct consequences of this mechanism qualitatively in the Penn Treebank in different ways: First, we notice that the probability of generating the <unk> token with our proposed network (VD-LSTM +REAL) is significantly lower compared to the baseline network (VD-LSTM) across many words. This could be explained by noting the fact that the <unk> token is an aggregated token rather than a specific word, and it is often not expected to be close to specific words in the word embedding space. We observe the same behavior with very frequent words such as "a", "an", and "the", owing to the same fact that they are not correlated with particular words. Second, we not only observe better probability assignments for the target words, but we also observe relatively higher probability weights associated with the words close to the targets. Sometimes this happens in the form of predicting words semantically close together which are plausible even when the target word is not successfully captured by the model. We provide a few examples from the PTB test set which compare the prediction performance of 1500 unit VD-LSTM and 1500 unit VD-LSTM +REAL in table 4. We would like to note that prediction performance of VD-LSTM +RE is similar to VD-LSTM +REAL for the large network.

---

[4]For PTB, small models were re-trained by initializing to their final configuration from the first training session. This did not change the final perplexity for baseline, but lead to improvements for the other models.

[5]Large network results on Wikitext-2 are not reported since computational resources were insufficient to run some of the configurations.

[6]This model was developed following our work in Inan & Khosravi (2016).

Table 4: Prediction for the next word by the baseline (VD-LSTM) and proposed (VD-LSTM +REAL) networks for a few example phrases in the PTB test set. Top 10 word predictions are sorted in descending probability, and are arranged in column-major format.

| **Phrase** + *Next word(s)* | Top 10 predicted words VD-LSTM | | Top 10 predicted words VD-LSTM +REAL | |
|---|---|---|---|---|
| information international said it believes that the complaints filed in + *federal court* | the 0.27 | an 0.03 | **federal 0.22** | connection 0.03 |
| | a 0.13 | august 0.01 | the 0.1 | august 0.03 |
| | **federal 0.13** | new 0.01 | a 0.08 | july 0.03 |
| | N 0.09 | response 0.01 | N 0.06 | an 0.03 |
| | ⟨unk⟩ 0.05 | connection 0.01 | state 0.04 | september 0.03 |
| oil company refineries ran flat out to prepare for a robust holiday driving season in july and + *august* | the 0.09 | in 0.03 | **august 0.08** | a 0.03 |
| | N 0.08 | has 0.03 | N 0.05 | in 0.03 |
| | a 0.07 | is 0.02 | early 0.05 | that 0.02 |
| | ⟨unk⟩ 0.07 | will 0.02 | september 0.05 | ended 0.02 |
| | was 0.04 | its 0.02 | the 0.03 | its 0.02 |
| southmark said it plans to ⟨unk⟩ its ⟨unk⟩ to provide financial results as soon as its audit is + *completed* | the 0.06 | to 0.03 | expected 0.1 | a 0.03 |
| | ⟨unk⟩ 0.05 | likely 0.03 | **completed 0.04** | scheduled 0.03 |
| | a 0.05 | expected 0.03 | ⟨unk⟩ 0.03 | n't 0.03 |
| | in 0.04 | scheduled 0.01 | the 0.03 | due 0.02 |
| | n't 0.04 | **completed 0.01** | in 0.03 | to 0.01 |
| merieux said the government 's minister of industry science and + *technology* | ⟨unk⟩ 0.33 | industry 0.01 | ⟨unk⟩ 0.09 | industry 0.03 |
| | the 0.06 | commerce 0.01 | health 0.08 | business 0.02 |
| | a 0.01 | planning 0.01 | development 0.04 | telecomm. 0.02 |
| | other 0.01 | management 0.01 | the 0.04 | human 0.02 |
| | others 0.01 | mail 0.01 | a 0.03 | other 0.01 |

## 7 CONCLUSION

In this work, we introduced a novel loss framework for language modeling. Particularly, we showed that the metric encoded into the space of word embeddings could be used to generate a more informed data distribution than the one-hot targets, and that additionally training against this distribution improves learning. We also showed theoretically that this approach lends itself to a second improvement, which is simply reusing the input embedding matrix in the output projection layer. This has an additional benefit of reducing the number of trainable variables in the model. We empirically validated the theoretical link, and verified that both proposed changes do in fact belong to the same framework. In our experiments on the Penn Treebank corpus and Wikitext-2, we showed that our framework outperforms the conventional one, and that even the simple modification of reusing the word embedding in the output projection layer is sufficient for large networks.

The improvements achieved by our framework are not unique to vanilla language modeling, and are readily applicable to other tasks which utilize language models such as neural machine translation, speech recognition, and text summarization. This could lead to significant improvements in such models especially with large vocabularies, with the additional benefit of greatly reducing the number of parameters to be trained.

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

# APPENDIX

## A  MODEL AND TRAINING DETAILS

We begin training with a learning rate of 1 and start decaying it with a constant rate after a certain epoch. This is 5, 10, and 1 for the small, medium, and large networks respectively. The decay rate is 0.9 for the small and medium networks, and 0.97 for the large network.

For both PTB and Wikitext-2 datasets, we unroll the network for 35 steps for backpropagation.

We use gradient clipping (Pascanu et al., 2013b); i.e. we rescale the gradients using the global norm if it exceeds a certain value. For both datasets, this is 5 for the small and the medium network, and 6 for the large network.

We use the dropout method introduced in Gal (2015); particularly, we use the same dropout mask for each example through the unrolled network. Differently from what was proposed in Gal (2015), we tie the dropout weights for hidden states further, and we use the same mask when they are propagated as states in the current layer and when they are used as inputs for the next layer. We don't use dropout in the input embedding layer, and we use the same dropout probability for inputs and hidden states. For PTB, dropout probabilities are 0.7, 0.5 and 0.35 for small, medium and large networks respectively. For Wikitext-2, probabilities are 0.8 for the small and 0.6 for the medium networks.

When training the networks with the augmented loss (AL), we use a temperature $\tau = 20$. We have empirically observed that setting $\alpha$, the weight of the augmented loss, according to $\alpha = \gamma\tau$ for all the networks works satisfactorily. We set $\gamma$ to values between 0.5 and 0.8 for the PTB dataset, and between 1.0 and 1.5 for the Wikitext-2 dataset. We would like to note that we have not observed sudden deteriorations in the performance with respect to moderate variations in either $\tau$ or $\alpha$.

## B  METRIC FOR CALCULATING SUBSPACE DISTANCES

In this section, we detail the metric used for computing the subspace distance between two matrices. The computed metric is closely related with the principle angles between subspaces, first defined in Jordan (1875).

Our aim is to compute a metric distance between two given matrices, $X$ and $Y$. We do this in three steps:

(1) Obtain two matrices with orthonormal columns, $U$ and $V$, such that span($U$)=span($X$) and span($V$)=span($Y$). $U$ and $V$ could be obtained with a QR decomposition.

(2) Calculate the projection of either one of $U$ and $V$ onto the other; e.g. do $S = UU^TV$, where $S$ is the projection of $V$ onto $U$. Then calculate the residual matrix as $R = V - S$.

(3) Let $\|.\|_{Fr}$ denote the frobenious norm, and let $C$ be the number of columns of $R$. Then the distance metric is found as $d$ where $d^2 = \frac{1}{C}\|R\|_{Fr}^2 = \frac{1}{C}\text{Trace}(R^TR)$.

We note that $d$ as calculated above is a valid metric up to the equivalence set of matrices which span the same column space, although we are not going to show it. Instead, we will mention some metric properties of $d$, and relate it to the principal angles between the subspaces. We first work out an expression for $d$:

$$
\begin{aligned}
Cd^2 = \mathrm{Trace}(R^T R) &= \mathrm{Trace}\left((V - UU^T V)^T (V - UU^T V)\right) \\
&= \mathrm{Trace}\left(V^T (I - UU^T)(I - UU^T)V\right) \\
&= \mathrm{Trace}\left(V^T (I - UU^T)V\right) \\
&= \mathrm{Trace}\left((I - UU^T)VV^T\right) \\
&= \mathrm{Trace}(V^T V) - \mathrm{Trace}\left(UU^T VV^T\right) \\
&= C - \mathrm{Trace}\left(UU^T VV^T\right) \\
&= C - \mathrm{Trace}\left((U^T V)^T (U^T V)\right) \\
&= C - \|U^T V\|_{Fr}^2 \\
&= \sum_{i=1}^{C} 1 - \rho_i^2,
\end{aligned}
\tag{B.1}
$$

where $\rho_i$ is the $i^{\text{th}}$ singular value of $U^T V$, commonly referred to as the $i^{\text{th}}$ principle angle between the subspaces of $X$ and $Y$, $\theta_i$. In above, we used the cyclic permutation property of the trace in the third and the fourth lines.

Since $d^2$ is $\frac{1}{C}\mathrm{Trace}(R^T R)$, it is always nonnegative, and it is only zero when the residual is zero, which is the case when span$(X) = $ span(Y). Further, it is symmetric between $U$ and $V$ due to the form of (B.1) (singular values of $V^T U$ and $V^T U$ are the same). Also, $d^2 = \frac{1}{C}\sum_{i=1}^{C} \sin^2(\theta_i)$, namely the average of the sines of the principle angles, which is a quantity between 0 and 1.

