# Peer review of "Tying Word Vectors and Word Classifiers: A Loss Framework for Language Modeling"

_ICLR 2017 — accepted_

[Public Comment · Andriy Mnih · 16 Nov 2016]
**Tying the input and output word representations used to be common**

I just wanted to point out that is used to be common to use the same embedding matrix for the input and target words in a neural language model. For example, the "cycling architecture" proposed in Yoshua Bengio's first language modelling paper [1] does that, as do all three models proposed in [2].

[1] Bengio, Y., Ducharme, R., Vincent, P. and Jauvin, C. A Neural Probabilistic Language Model. NIPS 2000
[2] Mnih, A., & Hinton, G. Three new graphical models for statistical language modelling. ICML 2007

[Official Review · AnonReviewer3 · rating 8 · confidence 4 · 15 Dec 2016]
**Nice argument for parameter sharing, promising results.**

This paper gives a theoretical motivation for tieing the word embedding and output projection matrices in RNN LMs. The argument uses an augmented loss function which spreads the output probability mass among words with close word-embedding. 

I see two main drawbacks from this framework:
The augmented loss function has no trainable parameters and is used for only for regularization. This is not expected to give gains with large enough datasets. 
The augmented loss is heavily “engineered” to produce the desired result of parameter tying. It’s not clear what happens if you try to relax it a bit, by adding parameters, or estimating y~ in a different way. 

Nevertheless the argument is very interesting, and clearly written.
The simulated results indeed validate the argument, and the PTB results seem promising.

Minor comments:
Section 3:
Can you clarify if y~ is conditioned on the t example or on the entire history.
Eq. 3.5: i is enumerated over V (not |V|)

[Official Review · AnonReviewer2 · rating 7 · confidence 4 · 16 Dec 2016]

This paper provides a theoretical framework for tying parameters between input word embeddings and output word representations in the softmax.
Experiments on PTB shows significant improvement.
The idea of sharing or tying weights between input and output word embeddings is not new (as noted by others in this thread), which I see as the main negative side of the paper. The proposed justification appears new to me though, and certainly interesting.
I was concerned that results are only given on one dataset, PTB, which is now kind of old in that literature. I'm glad the authors tried at least one more dataset, and I think it would be nice to find a way to include these results in the paper if accepted.
Have you considered using character or sub-word units in that context?

[Official Review · AnonReviewer1 · rating 6 · confidence 4 · 17 Dec 2016]
**Nice justification for parameter sharing**

This work offers a theoretical justification for reusing the input word embedding in the output projection layer. It does by proposing an additional loss that is designed to minimize the distance between the predictive distribution and an estimate of the true data distribution. This is a nice setup since it can effectively smooth over the labels given as input. However, the construction of the estimate of the true data distribution seems engineered to provide the weight tying justification in Eqs. 3.6 and 3.7.

It is not obvious why the projection matrix L in Eq 3.6 (let's rename it to L') should be the same as that in Eq. 2.1. For example, L' could be obtained through word2vec embeddings trained on a large dataset or it could be learned as an additional set of parameters. In the case that L' is a new learned matrix, it seems the result in Eq 4.5 is to use an independent matrix for the output projection layer, as is usually done.

The experimental results are good and provide support for the approximate derivation done in section 4, particularly the distance plots in figure 1.

Minor comments:
Third line in abstract: where model -> where the model
Second line in section 7: into space -> into the space
Shouldn't the RHS in Eq 3.5 be \sum \tilde{y_{t,i}}(\frac{\hat{y}_t}{\tilde{y_{t,i}}} - e_i) ?

[Final Decision · Program Chairs · 06 Feb 2017]
**ICLR committee final decision**

pros:
 - nice results on the tasks that justify acceptance of the paper
 
 cons:
 - In my opinion its a big stretch to describe this paper as a novel framework. The reasons for using the specific contrived augmented loss is based on the good results it produces. I view it more as regularization.
 - The "theoretical justification" for coupling of the input and output layers is based on the premise that the above regularization is the correct thing to do. Since that's really not justified by some kind of theory, I think its questionable to call this simple observation a theoretical justification.
 - Tying weights on the inputs and output layers is far from novel.